# A High-Density Polarized ³He Gas–Jet Target for Laser–Plasma Applications

Pavel Fedorets [1] , Chuan Zheng [1] , Ralf Engels [2] , Ilhan Engin [3] , Herbert Feilbach [4], Ulrich Giesen [5], Harald Glückler [5], Chrysovalantis Kannis [2] , Franz Klehr [5], Manfred Lennartz [5], Heinz Pfeifer [1], Johannes Pfennings [6], Claus Michael Schneider [1], Norbert Schnitzler [1], Helmut Soltner [5], Robert Swaczyna [4] and Markus Büscher [1,7,*]

1 Peter Grünberg Institute (PGI-6), Forschungszentrum Jülich GmbH, D 52425 Jülich, Germany; p.fedorets@fz-juelich.de (P.F.); c.zheng@fz-juelich.de (C.Z.); h.pfeifer@fz-juelich.de (H.P.); c.m.schneider@fz-juelich.de (C.M.S.); n.schnitzler@fz-juelich.de (N.S.)
2 Nuclear Physics Institute (IKP-2), Forschungszentrum Jülich GmbH, D 52425 Jülich, Germany; r.w.engels@fz-juelich.de (R.E.); c.kannis@fz-juelich.de (C.K.)
3 Sicherheit und Strahlenschutz (S-A), Forschungszentrum Jülich GmbH, D 52425 Jülich, Germany; i.engin@fz-juelich.de
4 Peter Grünberg Institute (PGI-JCNS-TA), Forschungszentrum Jülich GmbH, D 52425 Jülich, Germany; h.feilbach@fz-juelich.de (H.F.); r.swaczyna@fz-juelich.de (R.S.)
5 Central Institute for Engineering, Electronics and Analytics (ZEA-1), Forschungszentrum Jülich GmbH, D 52425 Jülich, Germany; u.giesen@fz-juelich.de (U.G.); h.glueckler@fz-juelich.de (H.G.); f.klehr@fz-juelich.de (F.K.); m.lennartz@fz-juelich.de (M.L.); h.soltner@fz-juelich.de (H.S.)
6 Institute of Biological Information Processing (IBI-TA), Forschungszentrum Jülich GmbH, D 52425 Jülich, Germany; j.pfennings@fz-juelich.de
7 Laser and Plasma Physics Institute, Heinrich Heine University, D 40225 Düsseldorf, Germany
* Correspondence: m.buescher@fz-juelich.de

**Abstract:** A laser-driven spin-polarized ³He²⁺-beam source for nuclear–physics experiments and for the investigation of polarized nuclear fusion demands a high-density polarized ³He gas-jet target. Such a target requires a magnetic system providing a permanent homogeneous holding field for the nuclear spins plus a set of coils for adjusting the orientation of the polarization. Starting from a transport vessel at a maximum pressure of 3 bar, the helium gas is compressed for a short time and can be injected into a laser–interaction chamber through a non-magnetic opening valve and nozzle, thus forming jets with densities of about a few $10^{19}$ cm⁻³ and widths of about 1 mm. The target comprises a 3D adjustment system for precise positioning of the jet relative to the laser focus. An auxiliary gas system provides remote target operation and flushing of the gas lines with Ar gas, which helps to reduce polarization losses. The design of the target, its operation procedures and first experimental results are presented.

**Keywords:** hyperpolarized ³He; gas-jet target; laser–plasma acceleration; polarized ion beams; phelix laser

**PACS:** 29.25.Lg,29.90.+r,52.38.Kd

## 1. Introduction

Nuclear polarized ³He gas has many applications in basic research and technology development. It can, for example, be used as a polarized neutron target [1] for studying the neutron structure in scattering experiments, e.g., with polarized electrons [2]. The development of compact polarized ³He-ion sources [3–6] provides additional options for the study of spin degrees of freedom in nuclear and particle physics. For laser-induced nuclear fusion reactions, spin-polarized fuel has the potential to provide a higher energy output. For example, calculations for the isospin-symmetric D(T,*n*)⁴He reaction with fully

polarized fuel predict an increase of the nuclear fusion cross-section by a factor of 1.5, while the energy gain increases by about 45% [7]. In case of the D-$^3$He reaction, a reduction of the required laser driver energy for ignition of about 60 % has been predicted as compared to unpolarized fusion [8].

A crucial question in the field of laser-induced particle acceleration is the influence of strong electro-magnetic laser and plasma fields on the spin polarization of the created particle beams. In simple words, two scenarios are conceivable: either the interaction times are long compared to the Larmor frequencies in the local fields and the spin orientation of a pre-polarized target is affected and destroyed, or the short (up to few ps) laser pulses (and the plasma) have only little effect on the spin alignment, and the polarization is conserved. For a detailed theoretical study of these effects, we refer to the paper by Thomas et al. [9]. Therefore, an experimental proof of nuclear spin–polarization conservation inside a (laser-induced) plasma is of high relevance also for fusion science.

In order to address the above-mentioned questions, experiments with nuclear polarized targets are needed. Such studies are very challenging since the energy differences of spin states in the strong plasma fields are of the order of meV only and, thus, much smaller than the typical thermal energies in the keV regime. As a first step of an experimental campaign, a measurement of proton distributions and their polarization, accelerated from ("standard") unpolarized foils, was performed at the Arcturus laser at Heinrich Heine University Düsseldorf, Germany [10]. As a second step, ion energy spectra and their angular distributions were measured with unpolarized $^{3,4}$He gas jets at the PHELIX laser, GSI Darmstadt, Germany [11,12]. Concluding experiments with polarized $^3$He gas require the development of a novel target system, which is the subject of this article (n.b.: $^4$He nuclei cannot be polarized, since they carry no spin).

## 2. Polarization and Relaxation of the Nuclear Spin

$^3$He gas has several advantages as compared to other polarizable materials. It can be rather easily polarized through optical pumping [13] and stored over a long time at room temperature in moderate (mT) holding fields. The polarization of the $^3$He gas decays exponentially with a relaxation time constant that is governed by several effects, i.e., the gradient of the magnetic field, dipolar relaxation, gas impurities, and surface relaxation on the walls of the storage vessel.

The homogeneity of the magnetic field is extremely important for achieving long relaxation time constants [14–18]. Therefore, all disturbing factors must be excluded or their influence at least minimized. All components in the vicinity of the polarized $^3$He gas should be made from non-magnetic materials, even if they are not in direct contact with the gas. Since all electro-magnetic excitations have to also be avoided, standard magnetic valves (e.g., driven by solenoids) must not be used in the gas lines.

Dipolar relaxation is the intrinsic polarization decrease due to magnetic dipole–dipole interactions between two $^3$He nuclei. This effect is thus proportional to the gas pressure inside the vessel and the gas lines [1,19]. Glass vessels with a maximal allowed pressure of 3 bar are typically used for the storage of polarized $^3$He. The relaxation-time constant is typically about 270 h.

An admixture of paramagnetic oxygen drastically decreases the relaxation time [20–23]. Therefore, careful cleaning, flushing, and pumping procedures are mandatory for the gas lines before the operation with polarized $^3$He. Surface relaxation is caused by the contact of the $^3$He nuclei with molecules of wall materials via adsorption and diffusion effects. According to Refs. [24,25], both effects are proportional to the surface-to-volume ratio of the gas vessel. Thus, spherical balloons produced at Mainz Univ. from Cs-layered glass can provide a relaxation-time constant in the range 430–570 h [24–26].

The standard 3 bar pressure of the polarized $^3$He gas in storage vessels is not sufficient for the experiments with laser-induced plasmas. Here, pressures in the range 20–30 bar are needed to drive the gas through the nozzle and to achieve the required densities at the laser–plasma interaction point [11]. Therefore, additional compression of the polarized

helium is required. Due to the increased depolarization, the duration of this compression should be as short as possible and be applied only to a minimal fraction of the polarized gas needed for the laser-induced ion-acceleration process.

## 3. Static Magnetic Holding Field

A constant homogeneous magnetic field is required to maintain the polarization of the $^3$He gas over sufficiently long time (i.e., many hours). This can, in principle, be achieved by two set-ups, Helmholtz coils, or permanent magnets. We chose permanent magnets due to specific conditions of future applications, like operation of the system in the vacuum of laser–interaction chambers and the presence of a huge *electromagnetic pulse (EMP)* from the laser–plasma interactions. Under such conditions, Helmholtz coils require a specific cooling system and the influence of EMPs on the magnetic field gradient is unclear. Clearly, the construction of the magnet system must fulfill the geometrical boundary conditions imposed by the vacuum chamber; in our case, it has been adapted to the one of PHELIX.

All these issues could be realized with a set of permanent magnets arranged in the geometry of a Halbach cylinder. In this geometry, each permanent magnet is oriented in such way to form a resulting common homogeneous field in the desired direction inside the cylinder. To be more specific, the magnet system consists of 48 NdFeB permanent magnets combined in eight vertical columns arranged in a circle with diameter 1100 mm [27,28]. The distance between the columns is 420.95 mm. Each NdFeB magnet has an octagonal cross-section with an energy product of 45 MGOe [29]. Figure 1 shows a sketch of the magnet system. There are six magnets in each column divided into two groups (2 × 3 magnets), with a vertical distance of 471.71 mm between the group centers (blue color). In this way, two rings of magnets are created. The resulting field is oriented horizontally, and, in the volume of the polarized $^3$He, its magnitude amounts to about 1.3 mT (at the center of the system). The estimated dominant relative field gradient in the horizontal direction is $(\delta B_z/\delta z)/B_0 \approx 3.5 \times 10^{-4}$ cm$^{-1}$, cf. Ref. [27]. The quality of the magnetic system was verified with a glass vessel containing polarized $^3$He located at the center of the system. Four calibrated flux-gate magnetometers [30] with a measurement range 0.1–200 nT were used pairwise [31–33] to observe any field changes during the operation. The measured relaxation time constant of polarized $^3$He inside the magnet system was 21 h [27]. This is sufficient for one working day at the planned experiments, where a fresh vessel with polarized $^3$He gas is used per day.

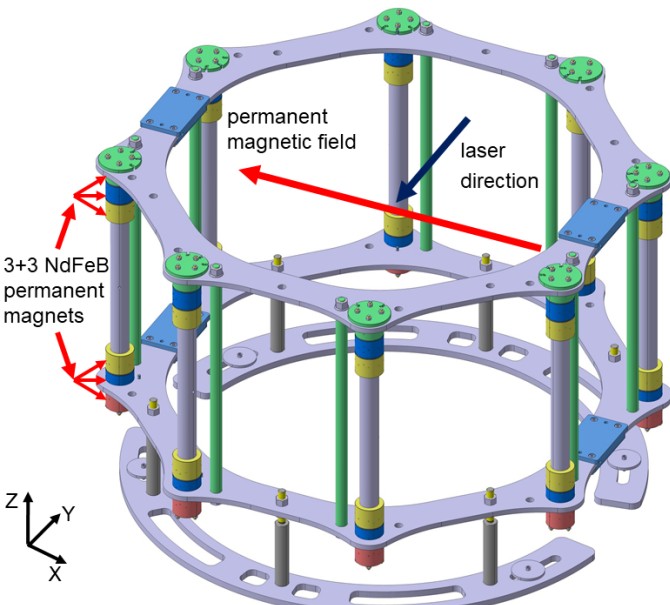

**Figure 1.** Layout of the permanent magnet system.

## 4. Coil System

In order to minimize the systematic error of the measured $^3$He$^{2+}$ beam polarization, it is mandatory to change the spin orientation of the prepolarized $^3$He gas for different laser shots relative to the laser-propagation direction. This is realized by a secondary magnet system composed of four concentric coils which provide a better field homogeneity than a Helmholtz set-up comprising just a pair of coils, see Figure 2. The field direction is also oriented horizontally but perpendicular to the field of the permanent magnets. The resulting magnetic field of both systems can thus be rotated in the horizontal plane by changing and/or reversing the current in the coils.

Each coil is made of a coiled Cu sheet with a cross section of $40 \times 40$ mm$^2$. The housing has a width and thickness of 56 mm, and the outer and inner diameters are 803 mm and 695 mm, respectively. The magnitude of the magnetic field at the center of the concentric coils system as function of the driving current was calibrated with a flux-gate magnetometer fine-adjusted perpendicular to the permanent magnetic field. The dependence is linear and described by $B = 503 \times I$, where $B$ (in µT) is the common magnetic field generated by the four concentric coils at the center of the system, and $I$ (in A) is the current applied to the coils. The planned current of 10 A corresponds to a field strength of 5.03 mT. In this case, the resulting magnetic field will be rotated by 75.5° relative to the original direction of the permanent magnetic field. Figure 3 shows the fully assembled magnet system in the laboratory.

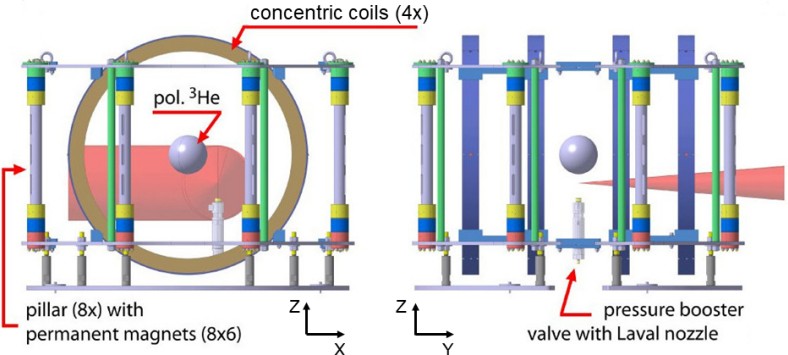

**Figure 2.** Arrangement of the magnet system with permanent magnets and four concentric coils. The transport vessel with $^3$He (gray sphere) can be seen at the center and the compressor below. The PHELIX laser beam is indicated by the red cone.

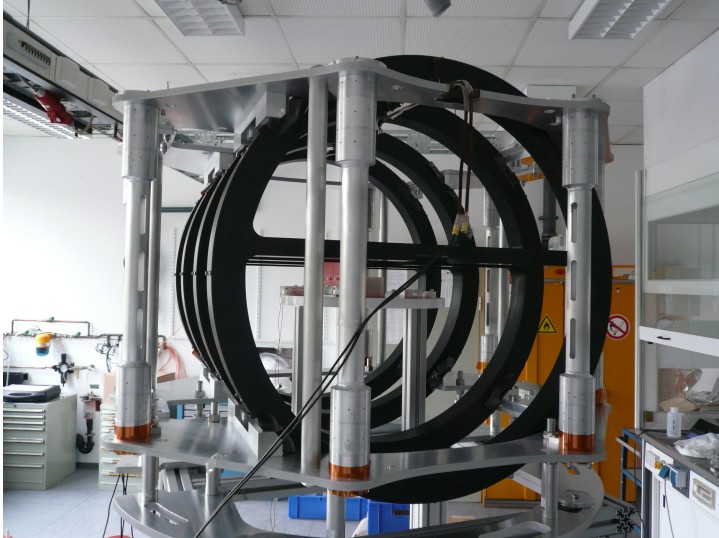

**Figure 3.** Fully assembled magnet system in the laboratory.

## 5. Generation and Transport of Polarized $^3$He Gas

The nuclear-polarized $^3$He gas is obtained by Metastable Optical Pumping (MEOP) [13], which is then stored in a ball-shaped glass vessel [18] shown in Figure 4. The glass material is GE 180. The diameter of the ball is 130 mm, which corresponds to a volume of 1.15 L. The glass ball is filled with $^3$He gas to a typical pressure of 3 bar. In order to preserve the polarization of $^3$He gas on the way from the polarizer (located in a different building of FZ Jülich) to the laboratory, the glass ball is transported inside a transport box [18] (see Figure 5), which again provides a homogeneous magnetic holding field induced by permanent magnets.

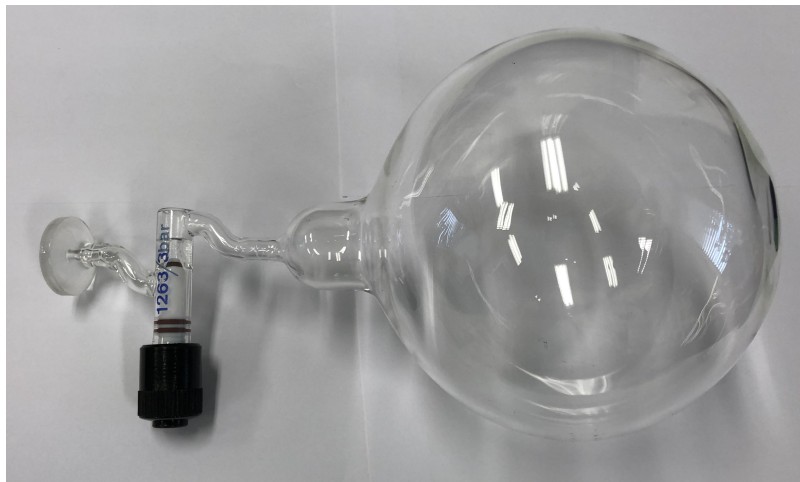

**Figure 4.** Glass ball, manufactured at FZ Jülich, for the storage of polarized $^3$He.

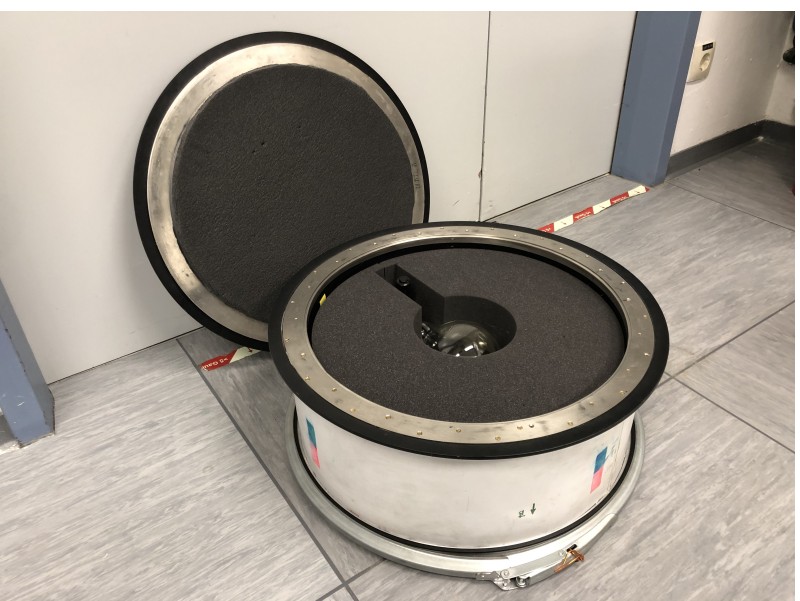

**Figure 5.** Magnetic transport box [18] with a single glass ball for polarized $^3$He gas. Alternatively, boxes are available that can accommodate three glass balls at the same time.

## 6. Gas Compressor with Nozzle

The pressure of the delivered polarized $^3$He gas from the polarizer is limited to 3 bar as an upper limit in the transport vessel. On the other hand, for the PHELIX experiments, a typical maximum particle density of a few $10^{19}$ cm$^{-3}$ at the interaction point with the laser pulse is required [11]. This value depends on several parameters, like the backing pressure of the gas in front of the nozzle exit, the shape and the minimal diameter of the

nozzle, and the distance between the nozzle exit and the laser focus. For other parameters given (see below), the pressure in front of the nozzle should be in the range of 18–27 bar. This requires an additional compression of the polarized $^3$He gas for a short time duration, directly before the laser shot. The completely non-magnetic compressor shown in Figure 6 was developed and built for this particular application.

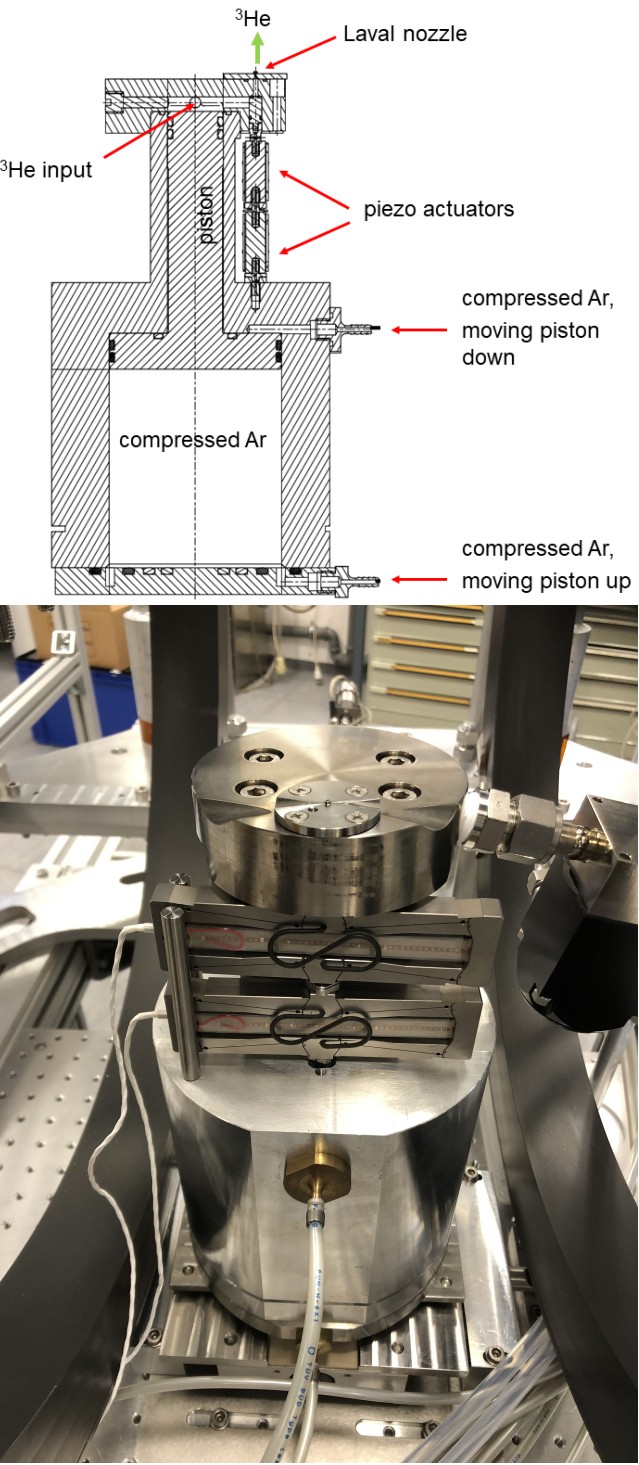

**Figure 6.** Top: Drawing of the compressor unit (edited figure from Ref. [34]). The piston is in the upper position, i.e., the $^3$He gas is compressed. Bottom: Photo of the compressor unit with the nozzle mounted on the top.

One of the main requirements is the use of non-magnetic materials for such a compressor device. Any magnetizable material—Even stainless steel—Will cause an instantaneous reduction of the $^3$He polarization. While in the past a similar compressor had completely been built from aluminium [35], in our case, a composite design from titanium and aluminium was chosen due to the required higher pressures. Titanium was used for the compression chamber (top part) as the volume with the high pressure and for the piston. The housing of the compressor is made from (cheaper) aluminium.

For sealing, O-rings made from the Flexible 80A Resin from Formlabs are used. The piston moves up and down by a pneumatic gas pressure of 6 bar. This operation gas must not contain any oxygen in order to avoid depolarization of the $^3$He gas in case of leakages into the compressor i.e., into the polarized $^3$He gas. Therefore, standard pressurized air is prohibited and cheap industrial gases like nitrogen or argon can be used instead. Both gases were used during preparatory tests, but later argon was selected, since it is easily removed from the recycled $^3$He gas by cryogenic purification methods.

The upper part of the compressor is equipped with a pre-valve in the $^3$He input tube, and a fast valve with a nozzle at the exit (see below). The operation cycle starts with the lower position of the piston. The polarized $^3$He gas is filled into the compression chamber with a maximum pressure of 3 bar. Under realistic conditions, the input pressure is not higher than 2.5 bar due to losses by the filling of the input gas lines and their covolume. During subsequent compression cycles, the input pressure decreases step by step, due to the finite, residual amount of the $^3$He gas in the transport vessel. Figure 7 shows the decrease of the input pressure after each compression for two possible options: either with removal of the unused $^3$He gas from the lines between two fillings or without removal. The first scenario leads to a faster pressure reduction, but it is needed for most laser–acceleration applications. High-intensity lasers like PHELIX have a minimum time interval between two shots of 1.5 h. During this time, the polarized $^3$He gas in the lines would basically lose its polarization due to the contact with the material of the thin gas lines (bad surface-to-volume ratio and hence many collisions with the tube walls) and due to magnetic field inhomogeneities, since the lines are not located at the center of the magnet system with the glass ball, where the magnetic field conditions are best. Therefore, it is mandatory to evacuate unused gas from the lines and refill them again from the glass ball shortly before the next laser shot, which reduces the amount of usable gas. Without the evacuation, an input pressure of 1.5 bar is reached at the 8th compression, and with evacuation procedure, only five compressions are possible in total. The pneumatic (radial diaphragm) pre-valve—which is able to stay closed at pressures up to 50 bar—in the input line prevents back-flow of compressed gas into the vessel. Like all the screws needed for its mounting, it is also made from titanium to ensure polarizion conservation.

Short-pulse laser applications like ion acceleration require operation in vacuum. Therefore, it is necessary to provide a high-density gas jet for a short time only, just before the laser pulse, to minimize deterioration of the vacuum conditions. The valve should completely be opened within a few-ms time window to form a sharp density profile. In our case, conventional electromagnetic valves are prohibited since they would reduce or completely destroy the helium polarization due to magnetic field gradients and hydraulic or pneumatic valves would be too slow. Piezo elements may provide the fast opening of the valve; however, commercial standard piezo-driven valves work at much lower pressures (up to 12 bar). Therefore, a new fast opening valve was designed and built. A piezo actuator P-602 [36] with an adjustment range of 1 mm at a maximum frequency of 150 Hz was chosen. In order to increase this range up to 2 mm, two such actuators were combined.

Figure 8 shows the measured reaction time ($\sim$6.6 ms) of the piezo element. The piezo-valve piston is mounted inside the top of the compressor to avoid additional dead volume. The piston is made from titanium and has a cylindrical rubber sealing on top. The distance between the opening aperture of the valve and the nozzle has been minimized to a few mm. When the compressor is idle (most of the time of the experiment), the piezo elements have no applied voltage and, hence, the piston is down, and the valve is opened. Directly

before compression, a voltage is applied to the piezo elements which causes a displacement, the piston moves up and the valve is closed. The filling with the (polarized) $^3$He gas, the compression and the opening of the piezo valve are integrated into the common trigger sequence of the laser–shot procedure.

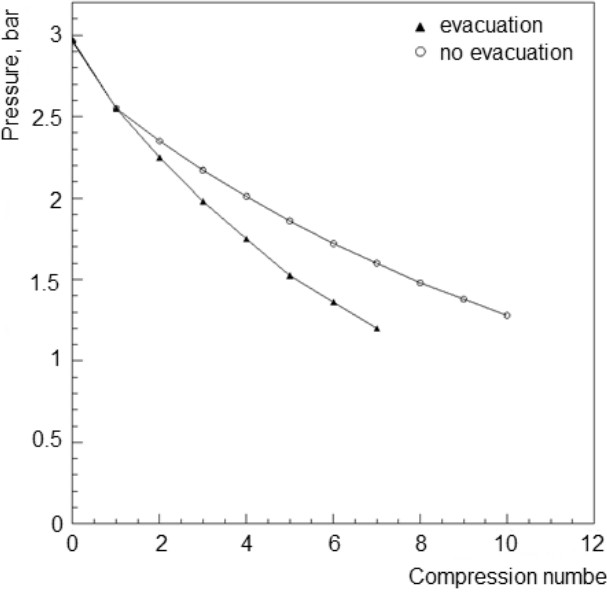

**Figure 7.** Decrease of the input pressure after each compression for two possible options. Step 0 corresponds to the initial pressure of 3 bar in the glass ball.

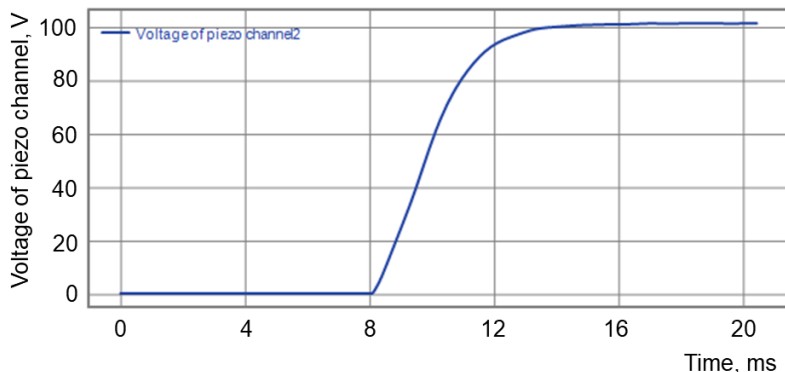

**Figure 8.** Time characteristics of the piezo element P-602 [36] used for the fast opening valve.

With the compressor piston in its lower position, the volume of the compression chamber is 90 mL. After compression, the volume is reduced to about 5.5 mL. The compression factor is expected to be 16.4 and for an input pressure of 3 bar a working pressure of 49 bar should be achievable. In reality, the connection with the pre-valve and the nozzle added some volume, which is difficult to calculate precisely. Therefore, a calibration procedure was carried out for the compressor, where the calibration was done with Ar as compressed gas and pressurized air at 6 bar as operation gas. Figure 9 presents the calibration function between the pressures of compressed and input gas. The drop at input pressures above 7 bar is due to backward flow in the pneumatic pre-valve. Short tests with a higher input pressure of 10 bar in the pneumatic line revealed recovery of the linear relation between input and output pressures.

The density distribution in the gas jet is one of the key parameters for laser–plasma experiments. Our previous measurements at PHELIX [11] revealed that this density should not fall below $4 \times 10^{18}$ cm$^{-3}$; otherwise, the number of laser-accelerated ions becomes too small for the polarimetry. For this reason, the useful gas-jet density was chosen to

be in the range of $(3–4) \times 10^{19}$ cm$^{-3}$. The particle density profile is basically determined by the nozzle geometry, a flat profile with sharp edges—which is most favorable for ion acceleration—can be generated by a supersonic de-Laval nozzle. Our nozzle with Mach numbers of $M_{super} \approx 3.44$ and $M_{sub} \approx 0.14$, a minimum diameter of 0.5 mm and nozzle exit of 1 mm, is made from titanium. The nozzle flange is mounted directly above the outlet of the piezo valve, see Figure 10.

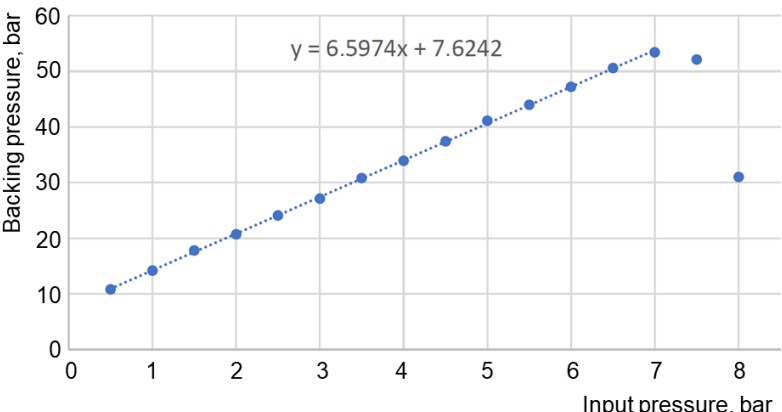

**Figure 9.** Dependence of the compressed gas pressure on the input gas pressure (edited figure from Ref. [34]).

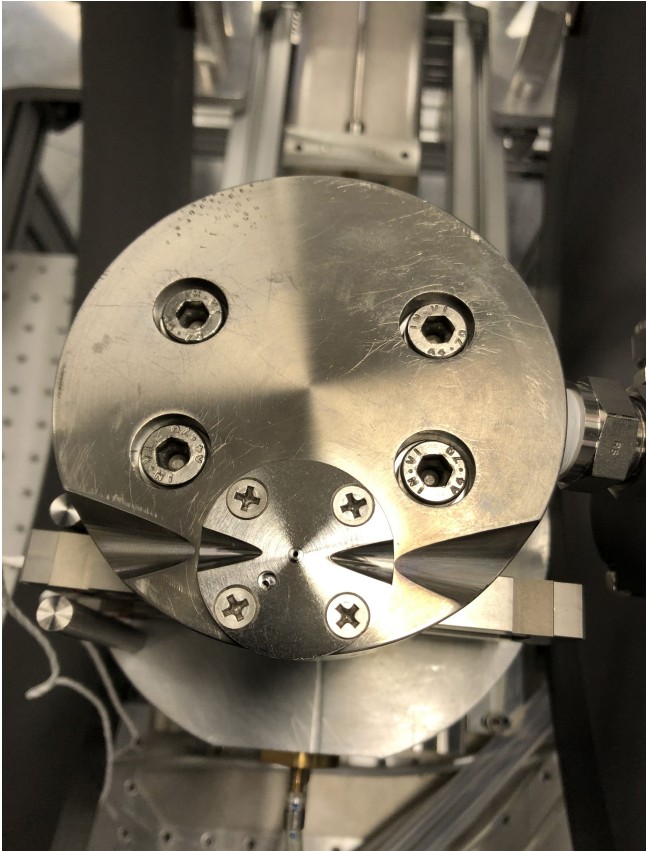

**Figure 10.** Nozzle mounted on the top of the compressor. The cone-shaped grooves follow the direction of the laser beam. At the lower left side of the nozzle, a needle for focus adjustment can be placed.

The density distribution of the gas jets was measured and calibrated with the help of a Mach–Zehnder interferometer (see Figure 11 in Ref. [37]). For these tests, the compressor

was placed in a vacuum chamber with glass windows for the laser beam, and the interferometric images were recorded with a CCD camera. Figure 12 shows examples of such images without and with the jet. The left picture presents the interferometric image above the nozzle (seen as black area in the lower part) before the trigger for opening the piezo valve on the compressor. The picture reveals a stable linear interferometric pattern that is parallel to the upper nozzle edge. The right picture was taken 32 ms after triggering the piezo valve, when the gas jet already reaches its density plateau. The image shows a visible distortion of the interferometric pattern, in particular at the central lower part close to the outlet from the nozzle. The abrupt change of the particle density is correlated with a change of the refractive index. As a result, the phase difference between the test and reference beams changes, which leads to a shift of the fringe pattern. The data analysis procedures and extraction of the density distributions are described in Ref. [37].

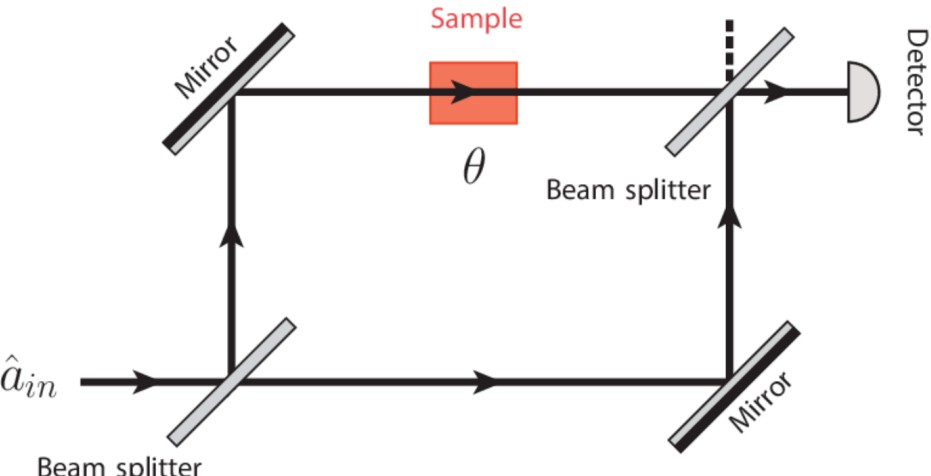

**Figure 11.** Sketch of the Mach–Zehnder interferometer for the measurement of the gas–density profile [37].

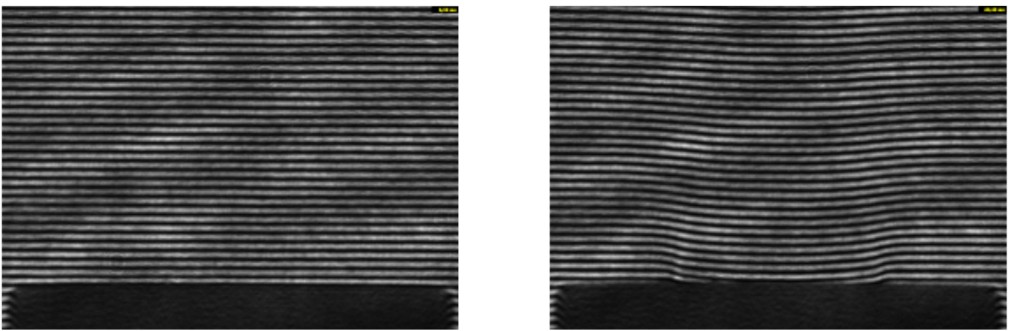

**Figure 12.** Interferometric images above the nozzle (black area at the lower edges) before gas-jet opening (**left**) and at 47.2 bar backing pressure 32 ms after triggering (**right**) [37].

Figure 13 shows a measured horizontal profile of the jet density 500 µm above the nozzle edge. The expected plateau shape is confirmed. The particle density depends linearly on the backing pressure (see Figure 14) and, thus, also on the $^3$He input pressure. The density of the jet decreases with the distance from the nozzle, see Figure 14. Figure 15 presents the temporal evolution of the particle density for different heights above the nozzle edge at 27.1 bar backing pressure. The gas jet reaches its maximum density at 14 ms after the trigger for opening the piezo valve [37] and remains stable for at least 30 ms after that. The main results of the interferometric measurements are summarized in Table 1. It is seen that the desired gas–jet density range of $(3–4) \times 10^{19}$ cm$^{-3}$ is achievable with the gas from the transport vessel, but the input pressure should not drop below 1.5 bar. This limits the operation with one glass ball to five compressions with evacuation of $^3$He gas from

the tubes between the laser shots. This is well suited for experiments at the PHELIX laser, which permits a maximum of six shots per day.

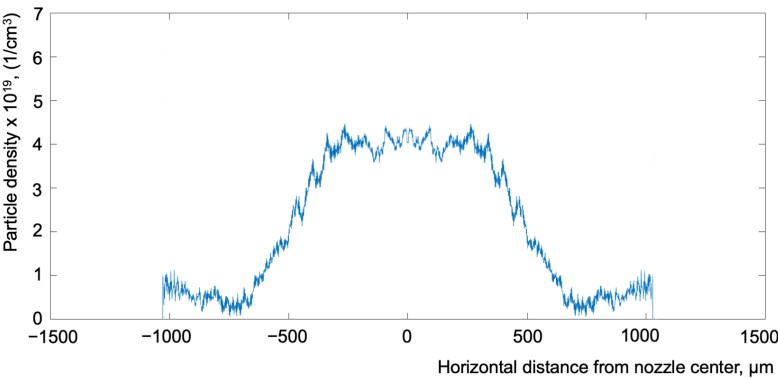

**Figure 13.** Horizontal distribution of the particle density at 27.1 bar and 500 μm above the nozzle edge, 32 ms after valve opening [38].

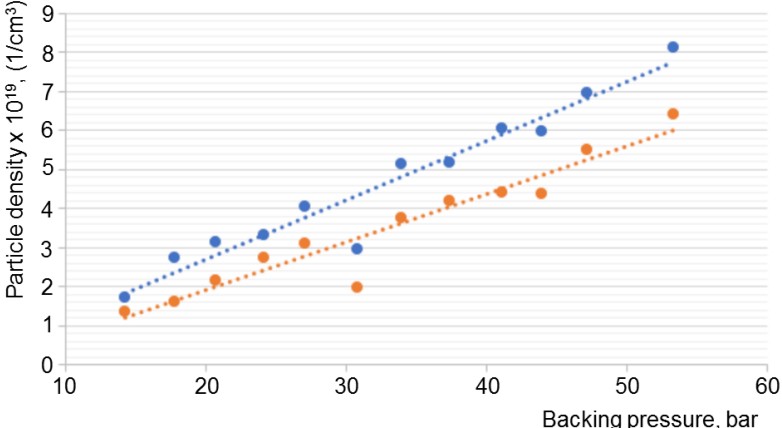

**Figure 14.** Maximum particle density for different backing pressure values between 17.8 and 53.4 bar [37]. The data are for 500 μm (blue dots) and for 1000 μm above the nozzle edge (brown dots).

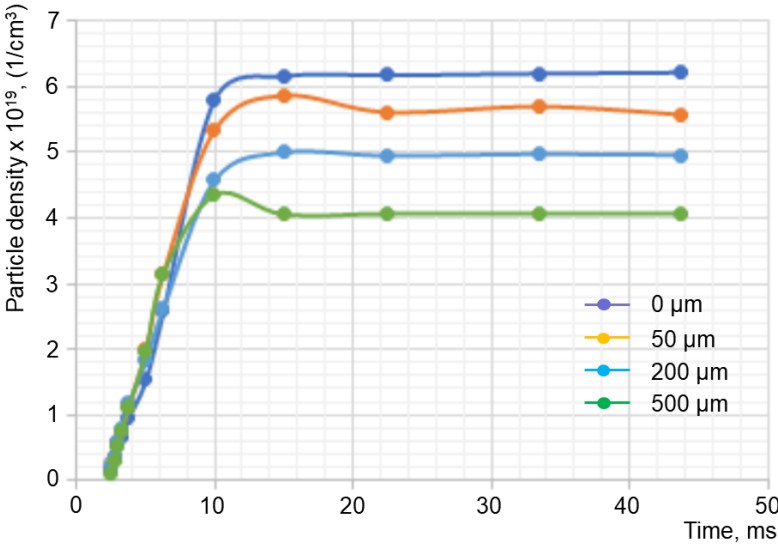

**Figure 15.** Time profile of the particle density for different heights above the nozzle edge at 27.1 bar backing pressure [37].

**Table 1.** Achievable gas densities at 500 μm and 1000 μm above the nozzle for various values of the input and backing pressures in the compressor [37].

| Initial Pressure (bar) | Backing Pressure (bar) | Particle Density at 500 μm (cm$^{-3}$) | Particle Density at 1000 μm (cm$^{-3}$) |
|---|---|---|---|
| 1 | 14.2 | $1.72 \times 10^{19}$ | $1.38 \times 10^{19}$ |
| 1.5 | 17.8 | $2.75 \times 10^{19}$ | $1.61 \times 10^{19}$ |
| 2 | 20.7 | $3.13 \times 10^{19}$ | $2.17 \times 10^{19}$ |
| 2.5 | 24.1 | $3.31 \times 10^{19}$ | $2.73 \times 10^{19}$ |
| 3 | 27.1 | $4.06 \times 10^{19}$ | $3.09 \times 10^{19}$ |
| 3.5 | 30.8 | $2.97 \times 10^{19}$ | $1.99 \times 10^{19}$ |
| 4 | 33.9 | $5.16 \times 10^{19}$ | $3.76 \times 10^{19}$ |
| 4.5 | 37.4 | $5.18 \times 10^{19}$ | $4.19 \times 10^{19}$ |
| 5 | 41.1 | $6.06 \times 10^{19}$ | $4.41 \times 10^{19}$ |
| 5.5 | 44.0 | $5.99 \times 10^{19}$ | $4.37 \times 10^{19}$ |
| 6 | 47.2 | $6.95 \times 10^{19}$ | $5.52 \times 10^{19}$ |
| 7 | 53.4 | $8.13 \times 10^{19}$ | $6.41 \times 10^{19}$ |

## 7. Adjustment System

For laser–plasma applications, such as ion acceleration, it is mandatory to adjust the gas jet (i.e., the upper nozzle edge) with sub-mm precision relative to the laser focus. The typical required accuracy is given by the diameter of the laser focus; at PHELIX, this amounts to 15–20 μm. In addition, the vertical separation of 0.5 mm between the laser beam axis and nozzle exit has to be controlled with high accuracy, since a smaller distance causes rapid degradation of the nozzle surface by the formed plasma and, consequently, of the jet quality. In addition, the gas jet must be precisely aligned with the collimator apertures of the $^3$He$^{2+}$ polarimeters, which are mounted at 90° relative to the laser propagation direction.

All of these issues were realized through the movement of the nozzle with the whole compressor unit, which has a mass of about 10 kg. The compressor unit rests on the adjustment system providing linear movements in any direction. This is provided by three linear actuators with stepper motors. Movement on the horizontal *X–Y* plane is achieved via linear actuators. The vertical displacement is realized by a horizontal actuator with a wedge connection. Figure 16 shows a drawing of the adjustment system.

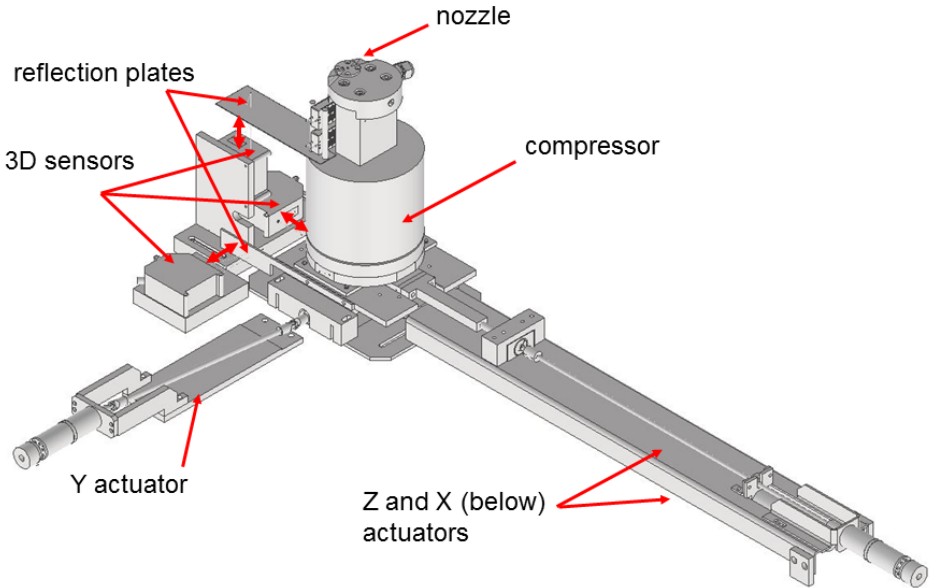

**Figure 16.** Adjustment system with the compressor.

A set of three distance laser sensors NCDT1700 was used for the control of the movement. A LabVIEW-based software receives the information from the sensors and sends the signals to the stepper motors. Independent movement of the compressor in all three directions is possible in step sizes from 1 μm up to 2 mm. The total range of the movement is ±20 mm in *X* and *Y*, and ±2.5 mm in the Z-direction.

The position of the nozzle was observed and verified with an additional diagnostics system based on two perpendicular (*X* and *Y* directions) lasers and images from CCD cameras. The fine positioning was performed with the image of a needle with a 50 μm tip. The needle was mounted on the top of the nozzle flange (see Figure 17).

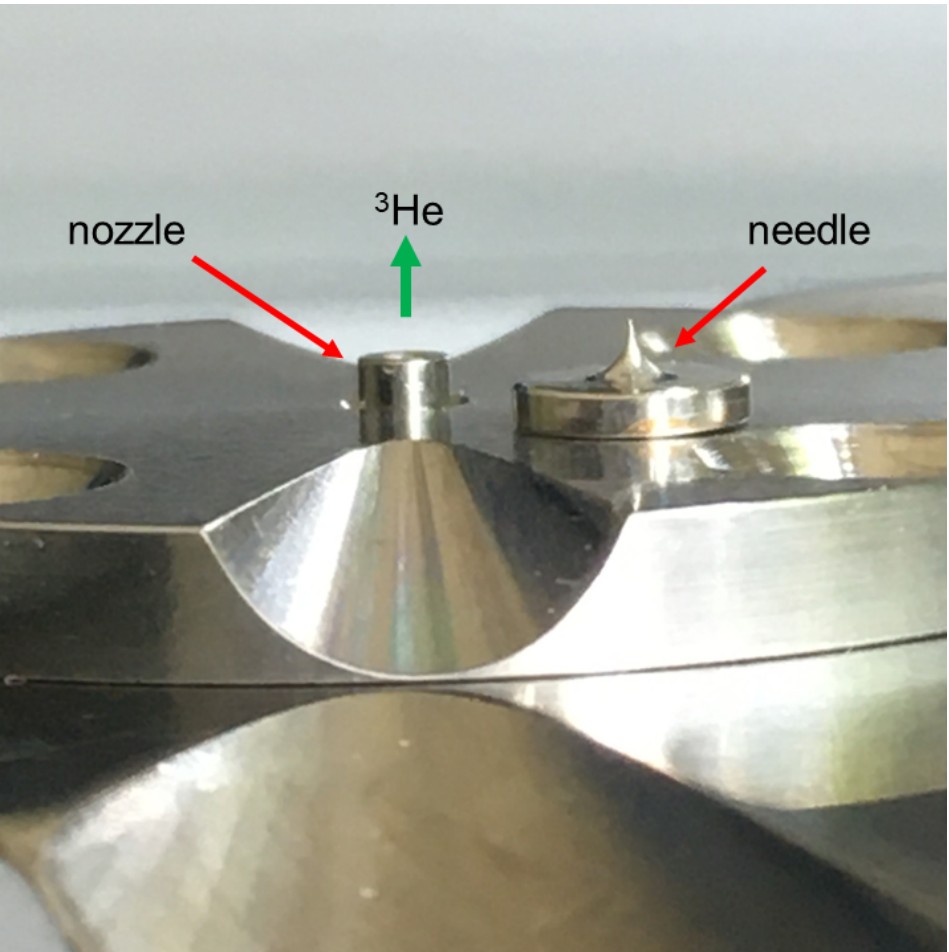

**Figure 17.** Nozzle with the needle for fine adjustment.

## 8. Auxiliary Gas System

For its use at a laser facility, the whole target must be ready for operation in a surrounding vacuum chamber. This includes the transfer of the polarized gas from the vessel to the compressor, as well as the supply of the operating Ar gas to the compressor. In addition, all components and pipes that are in contact with the polarized gas must be carefully cleaned shortly before target operation by repeated pumping and flushing with Ar. All this requires a system of auxiliary gas lines, vacuum pumps and valves.

Most critical are those valves which are in direct contact with the polarized gas. For their remote control inside a vacuum chamber, electromagnetic drivers are prohibited since they would distort the polarization. Therefore, tailored pneumatic valves produced from non-magnetic materials were designed and produced [39], which are operated by compressed Ar instead of air. Outside the vacuum chamber and far from the polarized gas commercial pneumatic Bürkert valves [40], types 5420 and 6013 are used for the pneumatic system and commercial hand valves [41] for the vacuum pipes.

Due to the high price of the 2200 EUR/liter, the $^3$He gas used needs to be collected in recycling tanks. For this purpose, a 50 L aluminum tank is connected to the output of a scroll pump. The evacuation of the $^3$He gas from the gas lines and its recycling should be made before each disconnection of the transport vessel from the gas system or any other opening of the system.

The operating gas system consists of several sub-units for $^3$He and Ar gas, vacuum lines, as well as recycling and pneumatic systems. A schematic overview can be found in Figure 18.

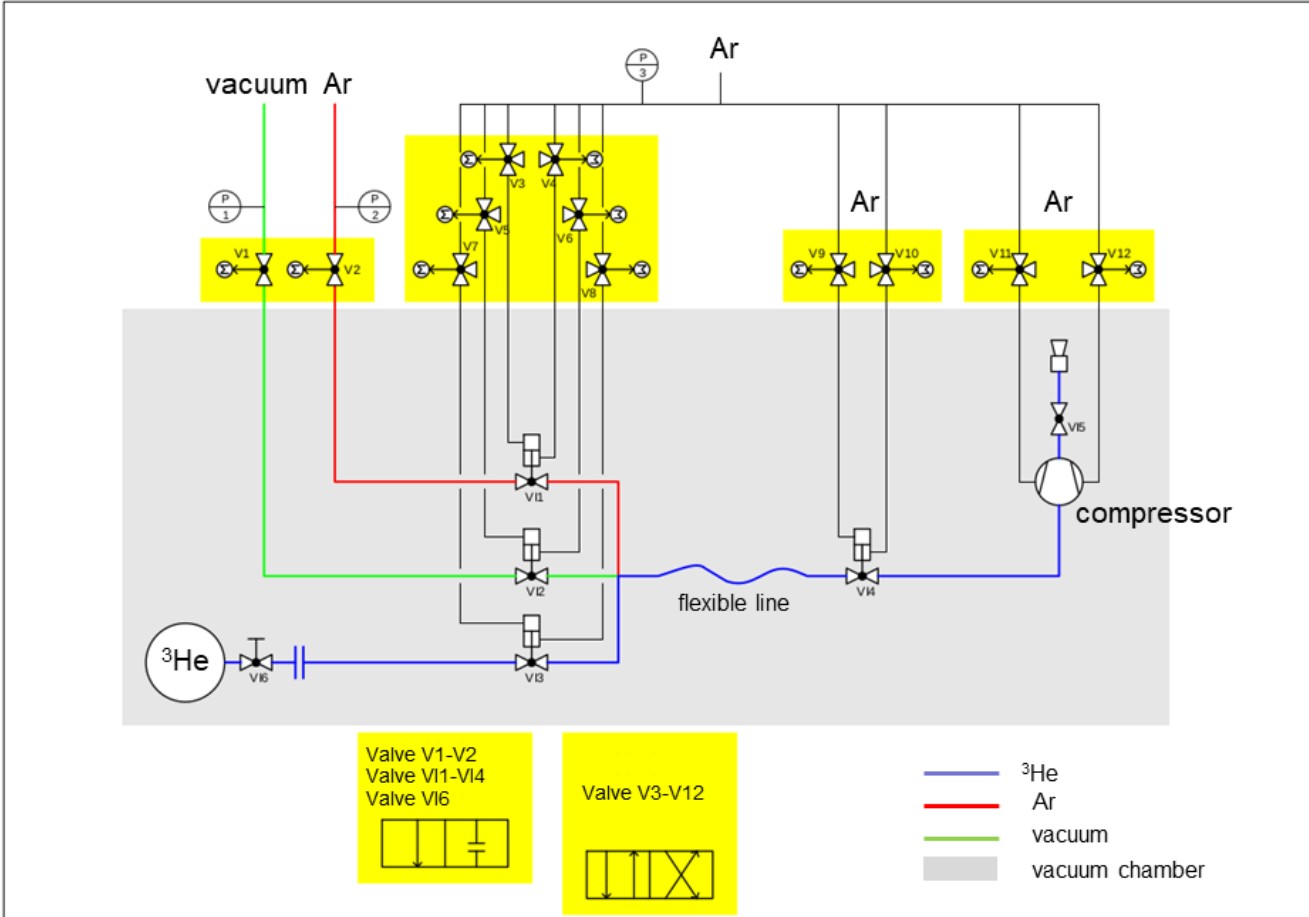

**Figure 18.** Schematic view of the gas system. The components that are located in the magnet system and inside the vacuum chamber are indicated by the gray rectangle. The $^3$He, Ar and vacuum lines are represented by different colors.

## 9. Control System

Remote control of the system is needed for operations inside the vacuum chamber and for implementation of a trigger commands sequence on an ms time scale for synchronization with the accelerating laser. The corresponding electronics is located inside a rack supplied with additional protection against the EMP signals from the laser–plasma interactions. The rack contains all the external pneumatic valves, a power-supply unit for the concentric coils, a power supply for the piezo units, trigger control unit and a computer with a touch screen for hand operation during preparatory procedures. The control software is based on LabView and consists of three units: for adjustment of the nozzle position, for applying the selected values of the current to the concentric coils and for operation of the pneumatic system with the compressor and the piezo valve. Figure 19 shows the rack with the control system as well as the magnetic system located inside a model of the laser vacuum chamber.

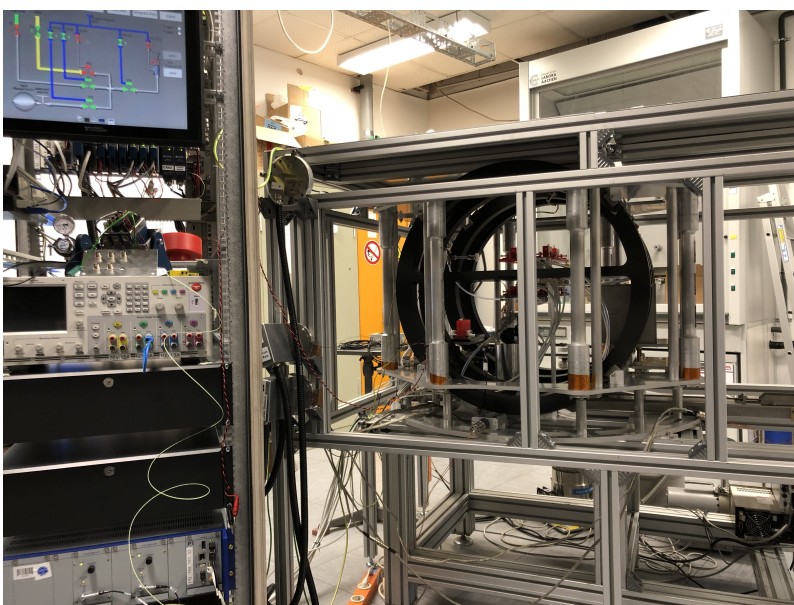

**Figure 19.** Rack with the control system (**left** side) and the magnetic system located inside a model of the laser vacuum chamber (**right**).

## 10. Test Operation with Two Glass Vessels

For commissioning, debugging and a full-scale test of the target system, a set of preparatory experiments has been carried out. The main goal was to study the factors influencing the polarization of the $^3$He gas in the jet after compression and vacuum injection. Measurements of the initial and final polarizations of the helium gas were achieved by the nuclear magnetic resonance (NMR) method. This equipment is a part of the helium polarizer. In order to perform these measurements, the used $^3$He gas was collected in a second identical glass vessel, which was also located inside the homogeneous magnetic field just above the input ball. The second ball was connected to the compressor through a short gas line with an adapter (instead of the nozzle) on top of the compressor. This test set-up is shown in Figure 20.

One of the first tasks was the search for a safe transfer procedure of the transport (input) vessel into the magnetic system without destroying the initial helium polarization. The problem is to avoid the zero-transition points of the magnetic field created by the Halbach cylinder geometry, since here the polarization would immediately disappear. The first option to achieve a safe transfer is to apply a high current to the concentric coils and to create a dominating strong magnetic field, which keeps its orientation far outside the magnet system without any zero transitions on the way from the transport box. In this case, it should be possible to insert the glass ball from the side along the field direction. Our tests show that such a field is possible to create, but smallest deviations from the central trajectory lead to an essential loss of polarization when passing through two neighbouring permanent magnets. Another option is the transfer of the glass vessel from above (through a flange in the lid of the PHELIX vacuum chamber). The Halbach arrangement of permanent magnets has no zero crossing along this direction. This idea was successfully confirmed during the tests.

As the next step, the preservation of the $^3$He gas polarization was ensured for the input vessel at its nominal position inside the holding field. For this test, the vessel was kept for one hour with a closed output valve and without contact of the gas with the gas lines. The following NMR measurement did not show any reduction of polarization.

The tests were then repeated with operation of the concentric coils in order to check the influence of changing magnetic field on the polarization losses. The current was smoothly increased up to 10 A, was kept constant for 10 min, and afterwards smoothly decreased back to zero. No influence on the degree of polarization was found.

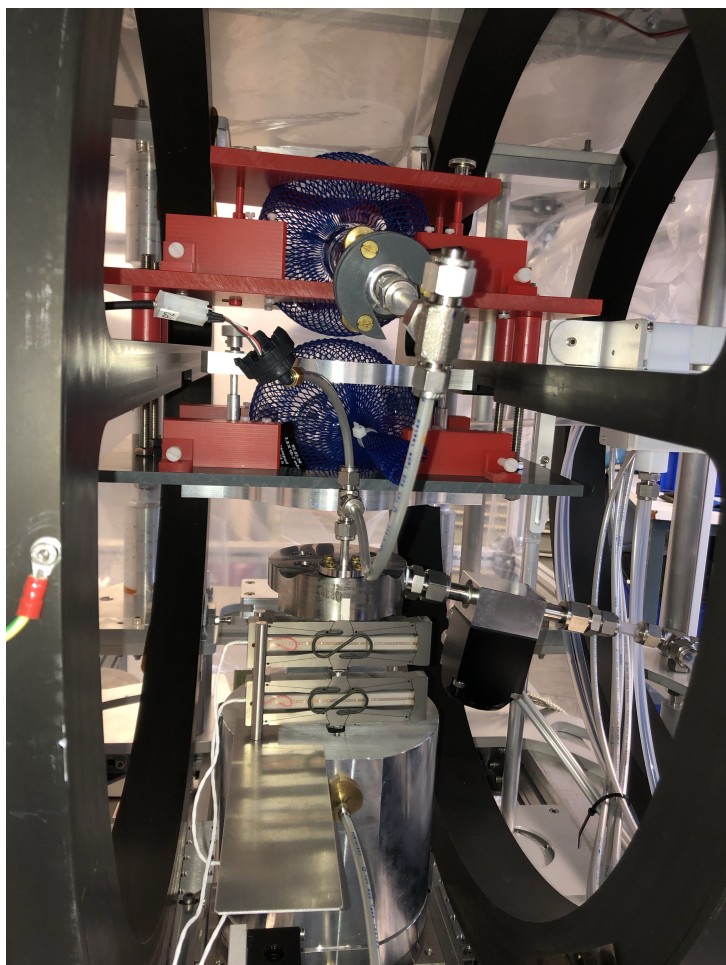

**Figure 20.** Test with polarized $^3$He. The input vessel with 3 bar is the lower one and the collection glass ball above.

After these initial studies, polarization conservation during complete compression cycles was tested. In each of these tests, nine compressions were carried out using polarized $^3$He gas from the input vessel with 3 bar. Each compression cycle includes the downward movement of the compressor piston, the filling of the compressor volume, the compression up to pressures of 18–24 bar, and the rapid opening of the piezo valve. Afterwards, the pressures in the input and the second glass ball are almost identical and amount to around 1.4 bar (some gas remains in the gas lines). Then, the polarization of the helium was measured in both vessels and compared. This relative measurement allows us to exclude any unnoticed loss effects during transport of the input vessel from the $^3$He polarizer. The tests reveal that the polarization is very sensitive to any contact of the $^3$He gas with various materials on the way from the transport to the second glass vessel. Even if these materials are not obviously magnetic (like stainless steel or brass), they can induce an essential loss of polarization. Most of the gas lines and valves thus consist of plastic, Teflon or titanium, but several (relatively small) elements are made from stainless steel. For a more detailed investigation of this effect, the transport glass ball was kept inside the magnetic system and connected to the gas lines with opened output valve. Thus, the polarized $^3$He gas was in contact with few of the stainless-steel elements without compression. Afterwards, the output valve was closed and the polarization inside the glass ball was measured by NMR. Most of the gas stayed in the glass ball, only 4–5% remained in the gas line. During a duration of 1 or 2 h, the polarization loss turned out to be in the range of 35–38% and does not depend on the gas pressure (1.6 or 3 bar).

After a whole series of further tests and implementation of improvements, a degree of $^3$He polarization after compression in the range 38–44% (compared to the remaining initial

gas in the input vessel) was achieved. It should be noted that these numbers were obtained with the Ti nozzle exchanged against an adapter from brass to conduct the polarized gas to the second glass ball. This additional component also contains a few stainless-steel elements and thus contributes to the reduction of polarization in the second glass ball. This effect—its extent is difficult to quantify—will be absent in laser–plasma experiments. Further studies to increase the degree of polarization in the gas jet are under way. One important detail is that another valve must be implemented directly behind the hand valve of the glass ball. The reason is that, before the target operation is started, the complete vacuum chamber must be pumped for hours and in this time a rather long part of the gas lines is open to the $^3$He gas reservoir. During this time, the volume-to-surface ratio is decreased and the polarization lifetime of the gas in the ball drops.

## 11. Conclusions

A target providing nuclear polarized $^3$He gas jets for laser–plasma applications has been designed, built and tested in the laboratory. Jet densities of $(3–4) \times 10^{19}$ cm$^{-3}$ at the laser–interaction point have been achieved. The measured degree of $^3$He polarization is sufficient for proof-of-principle polarization experiments at high-power laser facilities. First, such measurements have been carried out in August 2021 at the PHELIX facility, GSI Darmstadt. The data are still being analyzed, and the results will be topic of a separate publication. In the meantime, laboratory tests are continued with the goal to further increase the achievable degree of jet polarization. Another improvement can be the implementation of an NMR-system to measure the $^3$He polarization directly at the apparatus. For this purpose, the permanent homogeneous magnetic field of 1.5 mT would induce a Larmor frequency of about 30 kHz that can be easily detected.

**Author Contributions:** Conceptualization, I.E., F.K., M.L., H.S. and M.B.; data curation, C.Z.; funding acquisition, H.G., C.M.S. and M.B.; methodology, P.F., R.E., H.F., U.G., H.G., C.K., H.P., J.P., N.S., H.S. and R.S.; project administration, M.L. and M.B.; resources, H.F., M.L. and M.B.; writing-original draft preparation, P.F.; writing-review and editing, C.Z., R.E., I.E., H.G., C.M.S., H.S. and M.B. All authors have read and agreed to the published version of the manuscript.

**Funding:** This research received no external funding.

**Institutional Review Board Statement:** Not applicable

**Informed Consent Statement:** Not applicable

**Acknowledgments:** This work has been carried out in the framework of the *Ju*SPARC (Jülich Short-Pulse Particle and Radiation Center) project [42] and has been supported by the ATHENA consortium (Accelerator Technology HElmholtz iNfrAstructure) in the ARD programme (Accelerator Research and Development) of the Helmholtz Association of German Research Centres. Special thanks go to Werner Heil (retired Professor from Johannes Gutenberg-University Mainz) for providing the $^3$He polarizer and for answering many emergency calls, and to Vincent Bagnoud, Bernhard Zielbauer (PHELIX group) for valuable discussions.

**Conflicts of Interest:** The authors declare no conflict of interest.

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
