# Peer review of "A High-Density Polarized 3He Gas–Jet Target for Laser–Plasma Applications"

_instruments, doi:10.3390/instruments6020018_

Round 1

Reviewer 1 Report

The manuscript describes the design and first experimental results of a high-density 3He gas jet target and a magnetic system to provide a spin-polarized 3He source for laser-plasma experiments. This work is motivated by the potential application of polarized 3He2+ in nuclear-physics experiments, including polarized nuclear fusion.

The authors describe in great detail the different components of the setup including the different magnetic components, choice of materials to avoid depolarization, gas systems and so on, which clearly demonstrates the challenges associated with implementing this source. The authors can already point to previous publications on specific components of the setup and indicate that experimental results from laser-plasma interactions using this source will be published separately. 

The authors have been quite careful to provide detailed reasoning and descriptions. What seems to be largely missing is an indication of errors or fluctuations of the jet performance, e.g. shot-to-shot variation of the density or density profile. Understanding variations in the laser and plasma performance are key to deriving meaningful results from laser-plasma experiments. Characterizing the stability of the source is even more crucial in this scenario where the supply of 3He is limited. Therefore I make the following minor but important suggestions:
- Figure 9, 14, 15, Table 1: Add error bars to indicate shot-to-shot fluctuations.
- Figure 13: Either show multiple traces or a mean with shaded regions for the variation to indicate how stable the profile is.
- Explain how or if these variations impact laser-plasma experiments.

Other minor comments:
- Figure 1: It might be helpful to indicate explicitly as in Figure 2 where the magnets are.
- 4. Coil system (p.3): the authors outline section 3 that permanent magnets are more suitable than Helmholtz coils due to vacuum considerations, cooling and EMPs. Could the authors explain why the same issues do not apply in this scenario?
- Figure 14: there is a dip in particle density around 30 bar. Is there an explanation for this?
- 6. (line 227): what kind of laser was used for the interferometric measurement?

Most concerns are minor and I recommend this manuscript for publication after the authors added some detail on the stability/variations of their source.

Reviewer 2 Report

Polarised gas targets find many applications in scientific research and applications. This manuscript gives a very detailed information on how they developed and tested a high-density polarised 3He gas jet target for laser-plasma interactions, with key information on each sub-system such as magnetic field, coil, gas compressor, nozzle, adjustment system, control system etc. This is interesting to the communities which need polarised gas targets.

I have a few comments/suggestion and would like to discuss with the authors so as to improve this manuscript.

  1. For figure 1, can you insert a coordinate system indicating X, Y and Z clearly so that the readers can understand the directions more easily? In the text, for example, lines 93-101, the authors mentioned directions such as horizontal, vertical, Bz, z. With a coordinate system, this will be better understood.
  2. For the figure 2, a coordinate system can also be added to help readers understand the orientation.
  3. In figure 14, can you explain in the text what happens at location when backing pressure is 30 bars?
  4. In line 84, you used EMP there for the first time, it is better to use its full name where I think it is electromagnetic pulse?
  5. The quality of the figure 8 is not ideal for publication. The numbers on the axes can be made more clearly.
  6. In lines 230-232, the authors showed the density distribution via Mach-Zehnder interferometer and cited reference [42], I would suggest that the authors add a few more sentences to briefly explain the difference between these two plots.
  7.  Line 370, 3He can be changed to 3He.
  8. Overall, this manuscript is well prepared. However, the quality of some figures can be improved to meet the publication requirement, for example, figures 7, 8, 13 and 14.

Round 2

Reviewer 1 Report

The authors have responded sufficiently to reviewer comments. I recommend this manuscript for publication.

Reviewer 2 Report

The authors have made the requested improvements to the manuscript according to the comments from reviewers. The coordination system has been added to figures 1 and 2. Some typos have been corrected in the manuscript. A few more sentences have been added to explain the differences between two interferometric images.

The authors gave the possible reasons why there is a dip in particle density  at about 30 bar. I think these sentences can also be inserted in the texts even the reason was not found. 

Overall, I am happy about the current version of the manuscript.